# Osteogenic Differentiation of Mesenchymal Stem Cells with Silica-Coated Gold Nanoparticles for Bone Tissue Engineering

**DOI:** 10.3390/ijms20205135

**Published:** 2019-10-16

**Authors:** Chinnasamy Gandhimathi, Ying Jie Quek, Hariharan Ezhilarasu, Seeram Ramakrishna, Boon-Huat Bay, Dinesh Kumar Srinivasan

**Affiliations:** 1Temasek Life Sciences Laboratory, National University of Singapore, Singapore 117604, Singapore; gandhi@tll.org.sg; 2Lee Kong Chian School of Medicine, Nanyang Technological University, Singapore 636921, Singapore; quek0176@e.ntu.edu.sg; 3School of Biological Sciences, Nanyang Technological University, Singapore 637551, Singapore; 4Department of Anatomy, Yong Loo Lin School of Medicine, National University of Singapore, Singapore 117594, Singapore; anthe@nus.edu.sg (H.E.);; 5Department of Mechanical Engineering, Center for Nanofibers & Nanotechnology, Faculty of Engineering, National University of Singapore, Singapore 117576, Singapore; seeram@nus.edu.sg

**Keywords:** pcl, silk fibroin, silica-coated gold nanoparticles, nanofibrous scaffolds, mineralization, bone tissue engineering

## Abstract

Multifunctional nanofibrous scaffolds for effective bone tissue engineering (BTE) application must incorporate factors to promote neovascularization and tissue regeneration. In this study, silica-coated gold nanoparticles Au(SiO_2_) were tested for their ability to promote differentiation of human mesenchymal stem cells (hMSCs) into osteoblasts. Biocompatible poly-ε-caprolactone (PCL), PCL/silk fibroin (SF) and PCL/SF/Au(SiO_2_) loaded nanofibrous scaffolds were first fabricated by an electrospinning method. Electrospun nanofibrous scaffolds were characterized for fiber architecture, porosity, pore size distribution, fiber wettability and the relevant mechanical properties using field emission scanning electron microscopy (FESEM), porosimetry, determination of water contact angle, measurements by a surface analyzer and tabletop tensile-tester measurements. FESEM images of the scaffolds revealed beadless, porous, uniform fibers with diameters in the range of 164 ± 18.65 nm to 215 ± 32.12 nm and porosity of around 88–92% and pore size distribution around 1.45–2.35 µm. Following hMSCs were cultured on the composite scaffolds. Cell-scaffold interaction, morphology and proliferation of were analyzed by FESEM analysis, MTS (3-(4,5-dimethyl thiazol-2-yl)-5-(3-carboxymethoxyphenyl)-2-(4-sulfophenyl)-2H-tetrazolium inner salt) and CMFDA (5-choromethyl fluorescein acetate) dye assays. Osteogenic differentiation of MSCs into osteogenic cells were determined by alkaline phosphatase (ALP) activity, mineralization by alizarin red S (ARS) staining and osteocalcin expression by immunofluorescence staining. The results revealed that the addition of SF and Au(SiO_2_) to PCL scaffolds enhanced the mechanical strength, interconnecting porous structure and surface roughness of the scaffolds. This, in turn, led to successful osteogenic differentiation of hMSCs with improved cell adhesion, proliferation, differentiation, mineralization and expression of pro-osteogenic cellular proteins. This provides huge support for Au(SiO_2_) as a suitable material in BTE.

## 1. Introduction

Bone, which is an integral part of the skeleton, is known to be active throughout its lifetime as it undertakes structural remodeling in order to provide mechanical protection for internal organs. In addition, it facilitates locomotion and serves as a mineral storage system for calcium, magnesium and phosphate [1]. Bone has a high turnover rate as it undergoes continuous regeneration and remodeling during the healing process. However, the healing process is often delayed by vascular necrosis, atrophic non-union and osteoporosis. When excessive loss of bone is incurred by conditions such as fractures, accidents, age-associated degeneration, and post-tumor removal, bone grafting is required to stimulate healing and restore strength [2,3]. The graft act as a mold to fill in the gaps created by bone defects and this stimulates bone regeneration [4]. Globally, over 2.2 million people undergo bone grafting procedures each year for bone-related problems, which include both autologous and allogenic bone grafting. However, successful bone grafting is hampered by immune rejection, pathogen contamination and scarcity of available donor bone grafts due to a high medical demand [5]. These issues can be readily avoided with regenerative medicine using the patients’ own cells, and tissue engineering can serve as a platform. A biocompatible scaffold can potentially be fabricated to provide a suitable microenvironment for tissue regeneration by promoting cell proliferation, cell adhesion, differentiation, mineralization and extracellular matrix (ECM) deposition [6]. Bone tissue engineering (BTE) involves growing osteoprogenitor cells within a porous degradable matrix that mimics ECM to direct regeneration of bone at the defect site. As previously mentioned, BTE can provide an alternative solution to overcome the limitations of current clinically available treatments [7]. The composite scaffolds are biocompatible, biodegradable and osteo-inductive to eliminate any in vivo immunological reaction. They allow attachment of biomolecules, such as growth and angiogenic factors which promote cell function and stimulate tissue regeneration. The scaffolds will eventually degrade, leaving newly formed tissue to occupy the existing space [8,9]. Electrospinning is a versatile and cost-effective method that helps to generate such composite scaffolds. Micro and nano structural scaffolds with a high surface area to volume ratios, an interconnected porous structure and suitable mechanical properties can be fabricated using synthetic and natural polymer, thereby creating a fibrous matrix that closely mimics the ECM and provides support for the healing process [10]. Poly-ε-caprolactone (PCL), a synthetic polymer approved by the United States Food and Drug Administration (FDA), is renowned for its properties, such as biodegradability, biocompatibility, good mechanical stiffness, tissue-adaptability, and penetrability. However, PCL alone is insufficient for use as a scaffold material for BTE because it is hydrophobic, which makes it is not osteoconductive, and it also lacks integrin-binding sites for proper cell adhesion [11,12]. Thus, additional materials can be added to enhance the suitability of PCL for BTE. Silk fibroin (SF), another FDA approved naturally derived bioactive polymer, has been traditionally used in biomedical applications. SF possesses great strength, toughness and elasticity which makes it a good material option for use in the areas of controlled delivery, and as biomaterials and scaffolds for tissue engineering [13,14]. Silica is a vital substance for the formation of bone as it helps to improve the function of osteoblasts, inhibits the function of osteoclasts and promote mineralization by precipitating calcium phosphate in its early stages [14]. Furthermore, silica regulates the interaction between collagen and proteoglycans, which improves the quality of ECM [15,16,17]. Most importantly, silica can induce differentiation of stem cells into osteoblasts [18]. Ganesh et al. have demonstrated that silica nanoparticles are able to enhance strength and biological activity when incorporated into PCL scaffolds [11]. Gold nanoparticles (AuNPs) have also been widely used in the fields of diagnosis, targeted drug distribution and regenerative medicine [19,20,21]. AuNPs are appropriate agents for functionalization of electrospun scaffolds for bone regeneration, due to their properties of encouraging osteogenic differentiation in stem cells and osteoclast reticence [12,22,23,24]. They are water-soluble and hold desirable features, such as biocompatibility and ease of synthesis [25]. Hence, silica-coated gold nanoparticles are potentially valuable materials for biomedical applications, such as biolabeling, biosensing, medicinal diagnostics and drug delivery [26]. In the current study, PCL, PCL/SF and PCL/SF/Au(SiO_2_) composite nanofibrous scaffolds were fabricated by electrospinning and then evaluated for their capability to support bone tissue regeneration of human mesenchymal stem cells (hMSCs) by observing cell morphology, cell proliferation, osteogenic differentiation and mineralization. MSCs are clonogenic cell types present in the bone marrow stroma that have the potential for multi-lineage differentiation into cell types, including mesoderm-type cells, such as osteoblasts, adipocytes, and chondrocytes. These cells are often used in clinical studies and tissue engineering applications because of their differentiating capability and case of isolation [27]. Therefore PCL/SF/Au(SiO_2_) scaffolds promise potential promoting bone tissue regeneration with ECM deposition and high levels of MSC amplification.

## 2. Results and Discussion

### 2.1. Characterization of Nanofibrous Scaffolds

Composite scaffolds, which are fabricated by electrospinning, are known to exhibit characteristics such as biodegradability, large surface area, high porosity with interconnected pore structure, and biocompatibility to support cell growth for tissue regeneration [28]. The surface morphology of electrospun nanofibrous scaffolds were analyzed with FESEM at an accelerating voltage of 10kV by sputter coating. Figure 1a–c shows the FESEM image of PCL, PCL/SF, PCL/SF/Au(SiO_2_) fabricated composite nanofibrous scaffolds, which are porous and uniformly connected beads-free fibrous structures under controlled conditions. Figure 1d and e shows the TEM image of depicting embedded silica-coated gold nanoparticles at the surface of the fibers in the PCL/SF/Au(SiO_2_) nanofibrous scaffold. Au(SiO_2_) nanoparticles were polydispersed some amount of particles placed among the interfiber spaces. The fiber diameter of the nanofibrous scaffolds was found to be between 164 ± 18.65 nm to 215 ± 32.12 nm (Table 1).

The frequency range of fiber diameters for PCL, PCL/SF and PCL/SF/Au(SiO_2_) nanofibrous scaffolds are 12, 12 and 14, respectively (Figure 2), which implies that the scaffolds possess ideal diameters for optimal tissue engineering. PCL scaffolds show an average fiber diameter of 215 ± 32.12 nm (Figure 2a), where it subsequently decreased upon addition of SF to 164 ± 18.65 nm (Figure 2b). This is possibly due to an increase in solution conductivity caused by the addition of bioactive SF [29]. Liverani et al., reported that incorporation of nanoparticles influences increase in fiber diameter Similarly, addition of Au(SiO_2_) into PCL/SF results in an increase in average fiber diameter of PCL/SF/Au(SiO_2_) nanofibrous scaffolds 172 ± 24.22 nm [30] (Figure 2c).

### 2.2. Surface Wettability

The surface wettability of a biomaterial is a primary consideration in tissue engineering as it affects the extent of cellular adhesion and consequently the distribution of cells within the nanofibers and the rate of tissue regeneration [31]. Generally, cellular adhesion is poor on hydrophobic surfaces but better on hydrophilic surfaces. Hydrophilic surfaces supports diffusion of immobilized biomolecules and cellular wastes from the nanofibers, which aids regular cell function. Figure 3 displays the water contact angle image of electrospun mats in which scaffolds possess values greater than 90° are considered as hydrophobic in character. PCL scaffolds were observed to possess a value more than 90° which is considered as hydrophobic, with a contact angle of 135.10 ± 3.3°. Upon blending of PCL with SF, the contact angle was reduced to 78 ± 3.1°, which could be ascribed to the OH− groups present in SF as they can form H bonds with H_2_O, facilitating increased surface wettability [32]. The addition of Au(SiO_2_) into PCL/SF nanofiber further decreased the contact angles to 47.90 ± 2.9°. These values indicate that PCL/SF/Au(SiO_2_) scaffolds are the most suitable for stimulating cell growth and supports studies which had proven that a contact angle value less than 80° is desirable for cell attachment and growth [33].

### 2.3. Porosity

Possessing an optimal porosity range is a prerequisite for of nanofibrous scaffolds for effective tissue engineering application. This is as an appropriate geometry for cellular attachment, proliferation, differentiation, spreading and infiltration of cells into the scaffold for restoring ECM, which occurs through the proper exchange of nutrients and oxygen, induction of angiogenesis and cell recruitment [34]. Any scaffold which provides 90% and above porosity is considered to be desirable for promoting regeneration of tissues or organs, since it can support better nutrient diffusion and cell-scaffold interactions [35]. As shown in Table 1, the porosities of PCL, PCL/SF and PCL/SF/Au(SiO_2_) scaffolds were 88 ± 4.3%, 92 ± 6.3% and 90 ± 7.5%, respectively. The PCL/SF and PCL/SF/Au(SiO_2_) scaffolds were able to provide more desirable porosities for bone tissue regeneration. The addition of Au(SiO_2_) led to the formation of fiber with increased porosity while PCL scaffolds have a slightly lower porosity of 88% below the optimal requirement of 90%. Interconnectivity of the pores and pore size are important factors in fabrication of scaffolds. The pore sizes of PCL, PCL/SF and PCL/SF/Au(SiO_2_) scaffolds are 1.45 µm, 2.12 µm and 2.35µm respectively. Scaffolds with larger pore sizes are more optimal for cell migration, ingrowth of cell into the scaffold, nutrient diffusion, removal of metabolic waste and vascularization. 

### 2.4. Mechanical Strength

Nanofibers are required to have sufficient mechanical strength in order to resist the stress generated during tissue neogenesis. Figure 4 shows the stress and strain curve of fabricated nanofibrous scaffolds PCL, PCL/SF and PCL/SF/Au(SiO_2_). In this study, the average ultimate tensile stress, ultimate tensile strain and Young’s modulus (which determine the mechanical property of nanofibers) were analyzed. A highly porous polymer fabricated scaffold will commonly exhibit early elastic behavior, whereby it bends before breakage [36]. PCL showed a maximum tensile stress of 7.63 MPa and a tensile strain 165%. But upon the incorporation of SF, the values of tensile stress increased to 11.67 MPa, whereas, the tensile strain reduced to 46.51%. This increase in mechanical strength upon immobilization of SF may be attributed to the presence of crystalline domain which makes SF flexible [37]. Mohammad et al. reported that upon the incorporation of inorganic nanoparticle with PLGA and PLGA/GEL, the mechanical property of the scaffold was augmented [38]. Kim et al., studied that incorporation of Au nanoparticles into PEO nanofibers increase the crystallinity of nanofibers [39]. Similarly, our data proved that incorporation of Au(SiO_2_) into PCL/SF scaffolds increased the tensile strength to 12.11 MPa, and this could be caused by an increase in crystallinity of the nanofiber.

### 2.5. Cell Proliferation

Tissue regeneration is known to occur by recruitment of MSCs to the site of injury, followed by differentiation to osteoblasts, which then deposit bone by intramembranous ossification. Cell proliferation plays an important role in both physiological and pathological activity. For instance, cell proliferation of stem cells is a key factor for the repair process in every regenerating organ [40,41]. When culture on scaffolds is able to result in an amplification of cell number, it implies that the scaffold has sufficiently mimicked native ECM and is able to support cellular growth and differentiation by establishing effective communication between the cells and scaffold [42]. Figure 5 shows the MTS assay results which determines the proliferation of hMSCs on TCP, PCL, PCL/SF and PCL/SF/Au(SiO_2_) nanofibrous scaffolds on day 7, 14, and 21. hMSCs grown on PCL scaffolds have a lower level of proliferation in comparison to PCL/SF and PCL/SF/Au(SiO_2_) scaffolds at all time points, which can be attributed to the absence of active binding sites in PCL. The cells that were grown on PCL/SF and PCL/SF/Au(SiO_2_) scaffolds showed significantly higher (*p* < 0.05) proliferation levels, compared to those grown on TCP and PCL scaffolds, due to the presence of bioactive SF and Au(SiO_2_) which increase the hydrophilicity of the scaffold for adhesion of cells. Sundaramurthi et al. has previously reported that mesoporous silica nanofibers support the enhanced proliferation of bone marrow derived MSCs for bone regeneration [33]. Similarly, our results revealed that Au(SiO_2_) loaded PCL/SF scaffolds enhanced the ability of hMSCs to proliferate as compared to TCP, PCL and PCL/SF scaffolds. Silica-coated gold nanoparticles incorporated on the surface provide the ligands essential for stimulating cell growth and tissue formation by mediating specific biological signals present during cellular processes. Our results revealed that the structural or chemical variation of the nanofibrous scaffold by addition of SF and Au(SiO_2_) could stimulate proliferation of hMSCs without inducing toxicity, therefore, leading to the development of a successful substitute for bone tissue regeneration.

### 2.6. Cell-Scaffold Interactions

Physical and chemical properties of fabricated biocomposite scaffolds are important for cell-scaffold communication, cell to cell interactions and biological cell signaling for cell proliferation and distribution of ECM proteins. Primary identification of osteogenic differentiation is indicated by ECM deposition arising from the interaction between hMSCs and the scaffolds. Figure 6 depicts the cell morphology and ECM deposition upon the interaction of hMSCs with the PCL, PCL/SF and PCL/SF/Au(SiO_2_) nanofibrous scaffolds. Cells distributed within the fabricated nanofibrous scaffolds exhibited extension of filopodia to adjacent cells (Figure 6c,d) as compared to cells on TCP and PCL scaffolds (Figure 6a,b). No bone matrix proteins (mineralization) were observed in PCL scaffold as compared to PCL/SF and PCL/SF/Au(SiO_2_) scaffolds. Li et al. has previously reported that secretion of bone matrix protein, primarily bioapatites, are in the form of globular accretions [43]. Similarly, a globular accretion by calcification was observed in the PCL/SF/Au(SiO_2_) scaffold. PCL/SF and PCL/SF/Au(SiO_2_) scaffold favor secretion of ECM minerals with deposition of large mineral clusters. In Figure 6d, ECM mineral secretion is indicated with arrows. Cells were observed to have migrated gradually into the nanofibrous scaffold and enhanced cell-to-cell interaction, as seen from the high density of the dark areas in scaffold loaded with Au(SiO_2_). Furthermore, the formation of filopodia and secretion of ECM minerals indicate that cell-scaffold interactions occur at highest levels in the PCL/SF/Au(SiO_2_) scaffold as compared to that in PCL and PCL/SF scaffolds even though cell morphology was relatively comparable across all scaffolds.

### 2.7. CMFDA (5-Chloromethylfluorescein Diacetate) Dye Assay

Interaction between seeded hMSCs with the scaffolds may disturb their viability due to harmful substances immobilized within the scaffolds. To analyze the synergetic effect of incorporated Au(SiO_2_) on the PCL/SF nanofibrous scaffold, CMFDA dye assay was performed. CMFDA have compounds that contain chloromethyl derivatives of the classification of active cells in vitro. Live cells will be detected by CMFDA dye as brightly fluorescent cells. Figure 7 shows the extent of CMFDA fluorescence staining in hMSCs seeded in fabricated scaffolds after 21days of cell culture. It can be observed that cells that were grown on TCP and PCL scaffolds showed elongated cell morphology (Figure 7a,b), while the cells grown on PCL/SF and PCL/SF/Au(SiO_2_) scaffolds exhibited varying degrees of cuboidal osteoblast-like cell morphology (Figure 7c,d) suggesting osteogenic differentiation. Wang et al. showed that biomimetic bone substitute of collagen/ SF induced osteogenic differentiation of bone marrow derived MSCs [44]. Our observed results also proved that PCL/SF scaffold influenced osteogenic differentiation of hMSCs, due to the presence of SF, a bioactive protein. It is evident from Figure 7d, that the PCL/SF/Au(SiO_2_) scaffold have the most number of osteoblast-like cells (possibly due to the synergetic effect of SF and Au(SiO_2_).

### 2.8. Alkaline Phosphatase (ALP) Activity

Alkaline phosphatase (ALP) is a primary phenotypic indicator secreted by osteoblasts. Upregulation of ALP occurs during early osteogenesis. Assessing the levels of ALP in the fabricated PCL, PCL/SF and PCL/SF/Au(SiO_2_) scaffolds will help to validate differentiation of hMSCs towards an osteogenic lineage. As observed in Figure 8, PCL/SF/Au(SiO_2_) and PCL/SF nanofibrous scaffolds show significantly increased (*p* < 0.05) level of ALP expression when compared to PCL and TCP nanofibrous scaffolds on day 14. However, the ALP activity was significantly increased (*p* < 0.05) in PCL/SF/Au(SiO_2_) scaffold when compared to all other scaffolds on day 21. This is because Au(SiO_2_) may induce in vitro osteogenic differentiation of precursor cells, as well as improve in vitro osteogenic formation. Zhou et al. reported that silica-coated nanoparticles stimulate the osteogenic differentiation of bone marrow MSCs in vitro concomitant with the upregulation of ALP activity [45]. The observed results revealed that immobilization of Au(SiO_2_) in ECM could actively trigger ALP activity, verifying that PCL/SF/Au(SiO_2_) scaffolds support enhanced ALP activity which induces osteogenic differentiation.

### 2.9. Alizarin Red S (ARS) Staining

Alizarin red S (ARS) staining helps to assess matrix mineralization which supports the production of ECM by deposition of calcium. Calcium deposition was determined by ARS assay. The capacity to deposit minerals is a marker for mature osteoblasts, which can be used to prove that the MSCs seeded the scaffolds differentiated and entered into the mineralization phase to deposit mineralized ECM. Figure 9a shows the quantitative determination of calcium mineralization. The PCL and TCP scaffold shows lower mineral expression as compared to PCL/SF and PCL/SF/Au(SiO_2_) scaffolds whereas PCL/SF/Au(SiO_2_) shows significantly higher (*p* < 0.05) deposition of minerals on day 21. Zhou et al. earlier reported that silica-coated nanoparticles stimulate osteogenic differentiation of bone marrow MSCs by higher mineralization and ALP activity [45]. Further calcium deposition was qualitatively analyzed, as shown in Figure 9b. The potential of mineral deposition is an indicator of matured osteoblast formation. Figure 9b showed that PCL/SF and PCL/SF/Au(SiO_2_) scaffolds had more calcium deposition when compared to TCP and PCL scaffolds on day 14 and 21.

ECM mineralization by osteogenically differentiated MSCs is one of the important factors for BTE. Our results suggest that incorporation of Au(SiO_2_) upregulate the mineralization process and yield more mineral deposition as observed in Figure 6 and Figure 8.

### 2.10. Expression of Osteocalcin (OCN)

Mineralization can be determined by measuring the level osteocalcin (OCN) expression, a bone-specific protein secreted by Osteoblasts, which are involved in bone formation. OCN plays a major role in mineralization, because it is rich in glutamic acid that binds strongly to Ca2+ [46]. Incorporation of SiO_2_ stimulates initial cell adhesion and osteogenic gene expression during osteogenic differentiation. After 21 days of cell culture, immunofluorescence analysis revealed the presence of hMSCs as indicated by the expression of MSCs specific marker protein, CD90 denoted with green fluorescence (Figure 10b,f,j,n) and the osteogenic lineage differentiated cells marker protein, OCN with red fluorescence (Figure 10c,g,k,o). The nuclei of the cells were counterstained with DAPI denoted by the blue fluorescence (Figure 10a,e,i,m). This result supports the presence of osteogenic differentiation of hMSCs as shown by the dual expression of both CD90 and OCN (Figure 10d,h,l,p). The results observed that the cell cultured on PCL/SF/Au(SiO_2_) scaffolds exhibit cuboidal morphology of osteoblast and OCN expression representing osteogenic differentiation when compared to all other scaffolds. This result proved that Au(SiO_2_) incorporation stimulates not only early cell proliferation but also osteogenic protein expression during osteogenic differentiation. In order to facilitate translational research for repairing a bone injury, more in vivo and clinical studies are needed to extensively test their performance. 

## 3. Materials and Methods

### 3.1. Materials

Human bone marrow derived mesenchymal stem cells (MSCs) were purchased from Lonza (Morristown, NJ, USA). Dulbecc’s modified Eagl’s medium (DMEM), Nutrient Mixture F-12 (HAM), fetal bovine serum (FBS), antibiotics and trypsin-EDTA were procured from GIBCO Invitrogen (Carlsbad, CA, USA). CellTiter 96® Aqueous one solution was obtained from Promega (Madison, Wisconsin, USA). Hexafluoro-isopropanol (HFIP), Alizarin Red S and cetylpyridinium chloride, were purchased from Sigma-Aldrich, Singapore.

### 3.2. Fabrication of Nanofibrous Scaffolds

PCL liquefied in HFIP at 10% *(w/v)*; PCL/SF at 80:20 *(w/w)* at a concentration of 10% and PCL/SF/Au(SiO_2_) was prepared 70:25:5 *(w/w)* at the concentration of 10% in HFIP. The polymers with the above weight percentage prepared in the solvent HFIP were kept for rotation at room temperature for PCL, SF and Au(SiO_2_) homogenization. Prepared PCL, PCL/SF and PCL/SF/Au(SiO_2_) solutions were loaded in a 5 mL syringe with 24 G needles attached. Subsequently, they were connected to a syringe pump at a constant flow rate of 1.4 mL/h and a high-voltage electrical power of 13–17 kV (Gamma High Voltage Research, Inc., Ormond Beach, FL, USA). The syringe containing polymer solution will generate nanofibers due to the voltage difference between the collector plate and syringe needle. Nanofibers obtained by electrospinning were gathered by a collector stage kept at a distance of 13–15cm from the tip of the syringe. For cell culture purposes, nanofibers were collected on 15mm cover slips and kept in a desiccator for 24h to allow evaporation of residual solvents from the scaffolds.

### 3.3. Characterization of Nanofibrous Scaffolds

The morphology of nanofibers was analyzed under a scanning electron microscope (FESEM, FEI-QUANTA 200F, Hillsboro, OR, USA) at an accelerating voltage of 10 kV after sputter coated with gold (JEOL JFC-1200 fie coater). Six fibers (*n* = 6) were selected at random from the SEM images on all scaffolds for calculation of fiber diameter and porosity of nanofibers. Scaffold diameter was analyzed using the image analysis software Image J (Image Java, National Institutes of Health, Bethesda, MD, USA). The mechanical property of nanofibrous scaffold was determined by a tabletop micro-tester (Instron 3345, Norwood, MA, USA). The pore size of the electrospun mat was analyzed by capillary flow porosimeter (Porous Materials Inc., Ithaca, New York, USA). The wettability of each scaffold type was observed by using the sessile water drop contact angle analysis system (AST products, Billerica, MA, USA). Transmission electron microscopy (TEM) analysis was performed with JEOL JEM-1230 (Massachusetts, USA)electron microscope operated at an accelerating voltage of 40 kV for the examination of nanoparticle size and space morphologies distribution. TEM grids were prepared by placing a drop of the sample in water dispersion on a carbon-coated copper grid and drying at room temperature (25 °C).

### 3.4. In-vitro Culture of Human Mesenchymal Stem Cells (hMSCs)

Human mesenchymal stem cells (hMSCs) were first cultured in a 75cm2 cell culture flask in complete DMEM/F-12 medium with 10% FBS and 1% antibiotic. hMSCs were incubated in a humidified environment of 37 °C and 5% CO_2_ for seven days, the medium was changed once every three days. At an approximate confluency of 80, the cells were detached from the flask with trypsin-EDTA and extracted by centrifugation for 5min (3000 rpm). Cell pellets were collected by discarding the supernatant. Cells were counted with a hemocytometer and trypan blue solution was used to count the cells. The electrospun nanofibrous mats attached to 15 mm circular coverslips were placed into the wells of 24 well plates. On top of the coverslips stainless steel rings were placed to avoid lifting of scaffolds from the coverslips. Scaffolds were first sterilized with UV light for 3 h. They were then washed with 70% ethanol for 30 min before thrice washing with PBS for 15 min to remove traces of residual HFIP. 1ml of complete medium was subsequently added to each scaffold and soaked overnight before the cells were seeded. The hMSCs were distributed on the electrospun mats at a seeding density of 7000 cells/well. Tissue culture polystyrene (TCP) was used as a control.

### 3.5. Cell Proliferation Assay

Cell proliferation on the scaffold was observed by colorimetric MTS assay (Cell titer 96 Aqueous one solution Promega, Madison, WI, USA). MTS assay measures cell proliferation rate based on reducing yellow tetrazolium salt into purple formazan crystals by dehydrogenase enzymes produced in the mitochondria by living cells. The aqueous soluble formazan dye displays absorbance at 490 nm, and the quantity of dye excreted is directly proportional to the number of viable cells. In this study, proliferations of cells were assessed on 7, 14 and 21 days. To calculate the proliferation on a specific day point (day 7, 14, 21), media from the wells of seeded cells were removed and washed with PBS to eliminate dead cells and residue. The scaffolds were then incubated for 3 h in MTS and pure media at 1:5 ratio in 5% CO_2_ incubator at 37 °C. After 3 h, excreted formazan dye from each well was pipetted into 96 well plates, and absorbance was measured at 490 nm using a microplate reader (Synergy H1 Microplate reader, BioTek, Bad Friedrichshall, Germany). 

### 3.6. CMFDA Staining

Cell morphology analysis was conducted using FESEM (FEI-QUANTA 200F, Hillsboro, OR, USA). After 21 days of cell culture on the scaffolds, the plates were washed with PBS and fixed with 3% glutaraldehyde for 4h. After incubation, nanofibrous scaffolds were washed with water and dehydrated with increasing concentration of ethanol for 10min (30%, 50%, 75%, 90% and 100% *v/v*), then hexamethyldisilazane was added to the scaffolds before they were kept overnight to dry in the fume hood. Subsequently, the morphology of cell was analyzed with FESEM at an accelerating voltage of 10 kV after coating with platinum. Besides FESEM imaging, live cell imaging can also be determined by CMFDA assay. CMFDA is a fluorescent dye comprising chloromethyl derivatives which actively stain the viable cells. Live cells have the ability to absorb the CMFDA compound and emit a bright fluorescence light in 2h. This occurs as CMFDA diffuses across the cell membrane and reacts with cytosolic esterase which results in the creation of a CMFDA derivative which is brightly fluorescent. Thereafter, glutathione S-transferase simplifies the derivative, which changes the cells to a cell-impairment state. The cells were first incubated with CMFDA compound containing dimethyl sulfoxide (DMSO) and DMEM/F-12 for 2h before the media was removed and images were taken using microscope.

### 3.7. ALP Activity

Osteogenic differentiation of hMSCs was followed by examining its alkaline phosphatase activity (ALP). ALP activity was calculated using an alkaline phosphate yellow liquid substrate system for ELISA (Sigma Life Sciences, St. Louis, MO, USA). ALP catalyzes the hydrolysis of p-nitrophenyl phosphate (pNPP), a colorless organic phosphate ester substrate into p-nitrophenol and phosphate, a yellow product. ALP activity was analyzed after 7, 14 and 21 days of cell culture. The scaffolds were washed with PBS to eliminate cellular waste and 400 μL of p-nitrophenyl phosphate was added to each well for a 30min incubation before 200 μL of 2 M NaOH was added to quench the reactions. The resulting yellow solution was aliquoted into a 96 well plate and the absorbance was measured at 405nm by a microplate reader (Synergy H1 Microplate reader, BioTek).

### 3.8. ARS Staining

Alizarin red S (ARS) was used to determine and quantify mineralization of differentiated osteogenic cells. After 14 and 21 days of cells seeded on to the scaffolds, the scaffolds were washed thrice with PBS before fixing with 70% ethanol for 1 h. The scaffolds were then washed with water and stained with ARS for 30min. The scaffolds were again washed thrice with distilled water. Images were captured using Leica CTR6000. The dye was eluted by incubating the scaffolds with 10% cetylpyridinium chloride for 1 h and its absorbance was measured at 540nm using a microplate reader (Synergy H1 Microplate reader, BioTek, Germany).

### 3.9. Immunofluorescence Staining

Osteogenic differentiation was analyzed using immunofluorescence staining which utilizes the MSCs specific marker protein CD90 and osteoblast-specific marker protein OCN. On day 21, the cells were first fixed with 100% methanol. The scaffolds were then washed with PBS and incubated in 0.1% Triton X-100 solution to permeabilize the cell membrane. The cells were then incubated in 3% bovine serum albumin for 1h to block non-specific binding sites. Subsequently, the primary antibodies were added to the cells for incubation of 1hr at room temperature. This was followed by adding the secondary antibody with incubation for 1h. The scaffolds were washed thrice with PBS and incubated with osteoblast-specific protein for 1h. After which, the secondary antibody was added and incubated for 1h. The scaffolds were washed with PBS to remove excess staining. Finally, the cells were stained with DAPI for 30min. The scaffolds were retrieved from well plates and mounted over the glass slide using Vectashield mounting medium and examined under the fluorescence microscope.

### 3.10. Statistical Analysis

Experiments were carried out in triplicates, and the data retrieved were expressed as mean ± standard deviation (SD). Statistical analysis was performed by One Way Analysis of Variance (ANOVA), and significance was determined at *p* < 0.05.

## 4. Conclusions

Electrospun hybrid PCL/SF/Au(SiO_2_) nanofibrous scaffolds have potential mechanical strength and biocompatibility similar to that of the bone matrix. The interconnecting porous structures of the scaffolds provided more structural space for the proliferation and mineralization of MSCs allowing sufficient exchange of nutrients and removal of waste products from metabolic processes. The findings in the present study show that the PCL/SF/Au(SiO_2_) nanofibrous scaffold has the most suitable surface among those tested for adhesion, proliferation, osteogenic differentiation and mineralization of hMSCs. Cells grown on PCL/SF/Au(SiO_2_) nanofibrous scaffolds showed high ALP levels with cuboidal osteoblast cell morphology and extracellular matrix mineralization along with the increased production of OCN in comparison to all other scaffolds. Therefore, Au(SiO_2_) loaded PCL/SF nanofibrous scaffolds can possibly contribute to an important role BTE, and consequently, bone treatment. Therapeutic uses of hMSCs cultured on PCL/SF/Au(SiO_2_) nanofibrous scaffolds hold great potential for the treatment of bone defects.

## Figures and Tables

**Figure 1 ijms-20-05135-f001:**
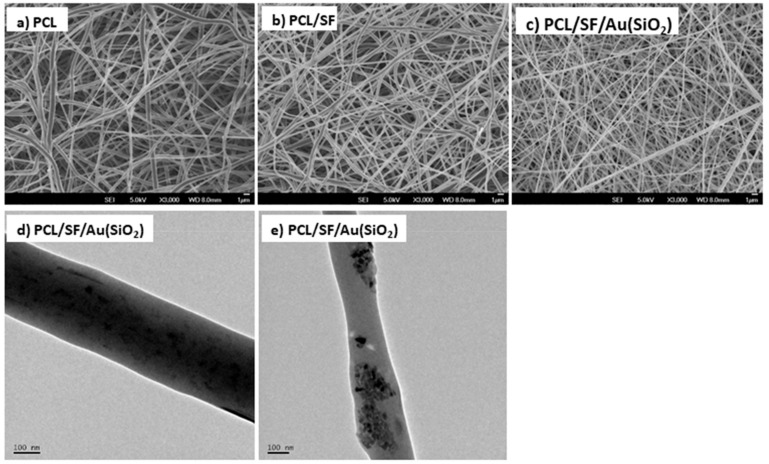
FESEM images of the electrospun (**a**) PCL, (**b**) PCL/SF, (**c**) PCL/SF/Au(SiO_2_) nanofibrous scaffolds. (**d**,**e**) TEM images of PCL/SF/Au(SiO_2_) nanofibrous scaffolds.

**Figure 2 ijms-20-05135-f002:**
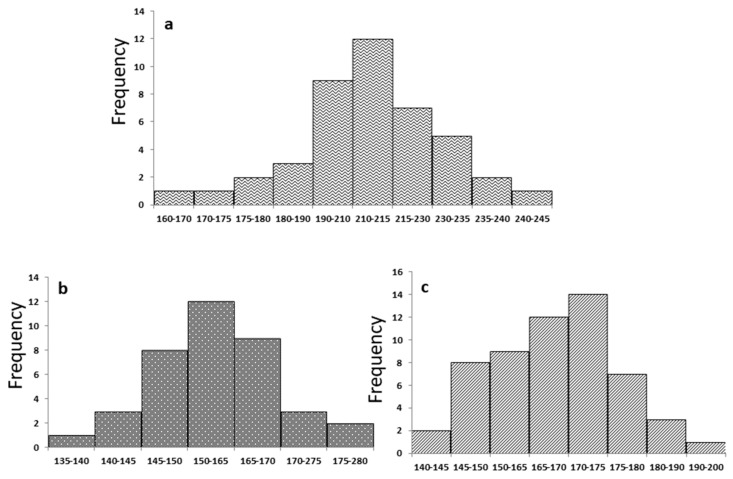
Frequency range of fiber diameters for (**a**) PCL, (**b**) PCL/SF, (**c**) PCL/SF/Au(SiO_2_) nanofibrous scaffolds.

**Figure 3 ijms-20-05135-f003:**
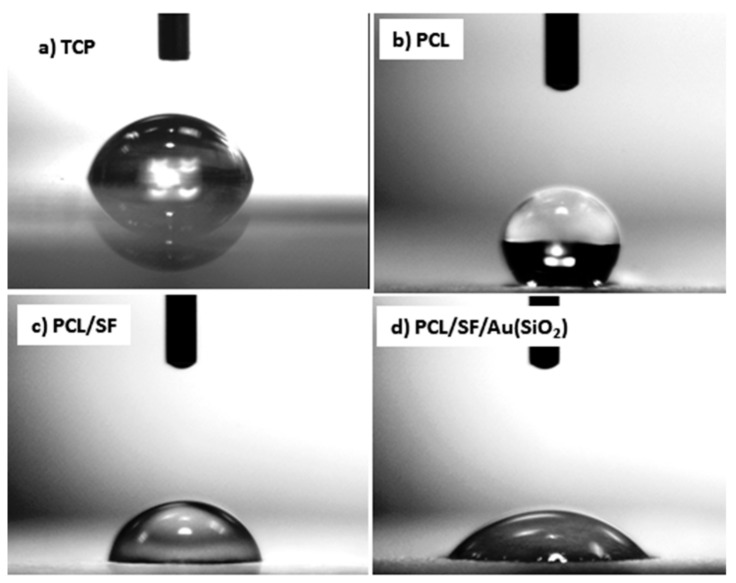
Water contact angle of (**a**) TCP (**b**) PCL (**c**) PCL/SF and (**d**) PCL/SF/Au(SiO_2_) nanofibers.

**Figure 4 ijms-20-05135-f004:**
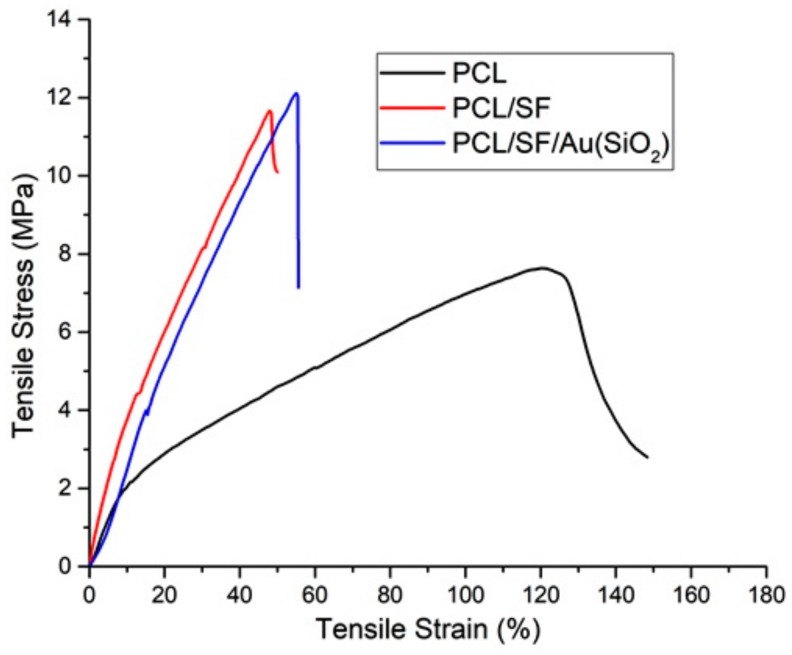
Tensile stress–strain curves of PCL, PCL/SF, PCL/SF/Au(SiO_2_) nanofibrous scaffolds.

**Figure 5 ijms-20-05135-f005:**
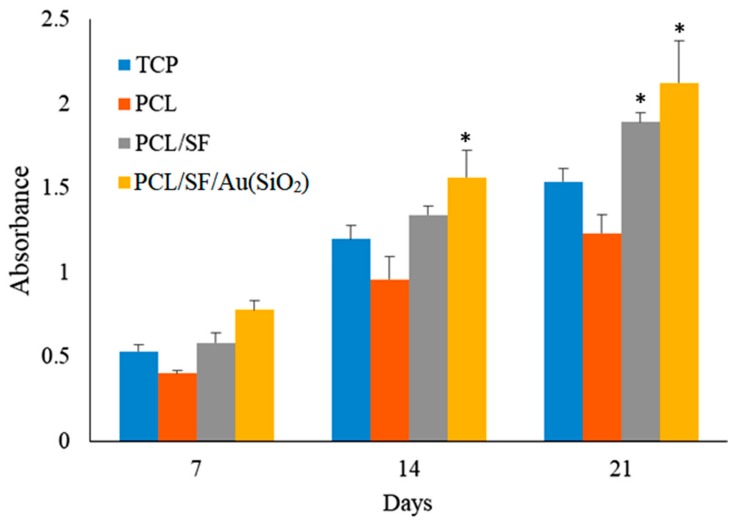
Cell proliferation of hMSCs on TCP, PCL, PCL/SF and PCL/SF/Au(SiO_2_) nanofibrous scaffolds on day 7, 14 and 21. * *p* < 0.05.

**Figure 6 ijms-20-05135-f006:**
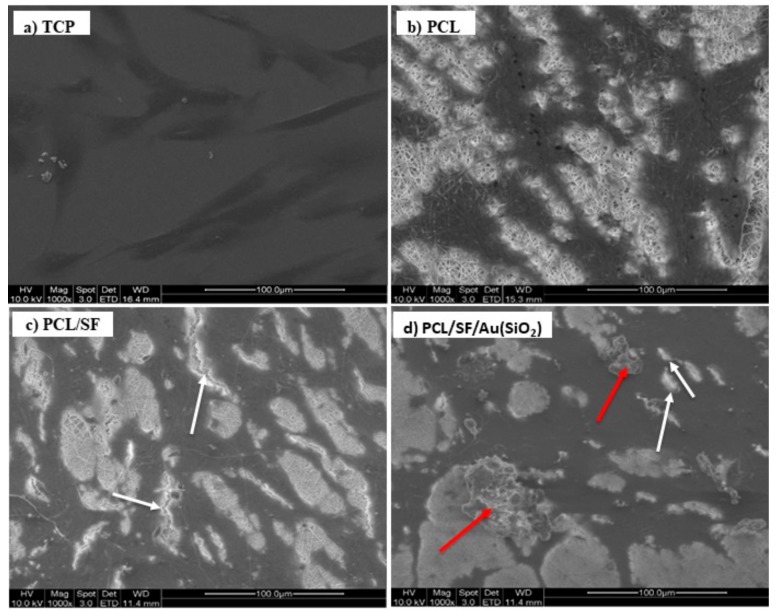
FESEM images showing the cell-biomaterial interactions on (**a**) TCP, (**b**) PCL (**c**) PCL/SF and (**d**) PCL/SF/Au(SiO_2_) nanofibrous scaffolds on day 21. Red arrows indicate the minerals secreted by hMSCs, while white arrows refer to the filopodia formed.

**Figure 7 ijms-20-05135-f007:**
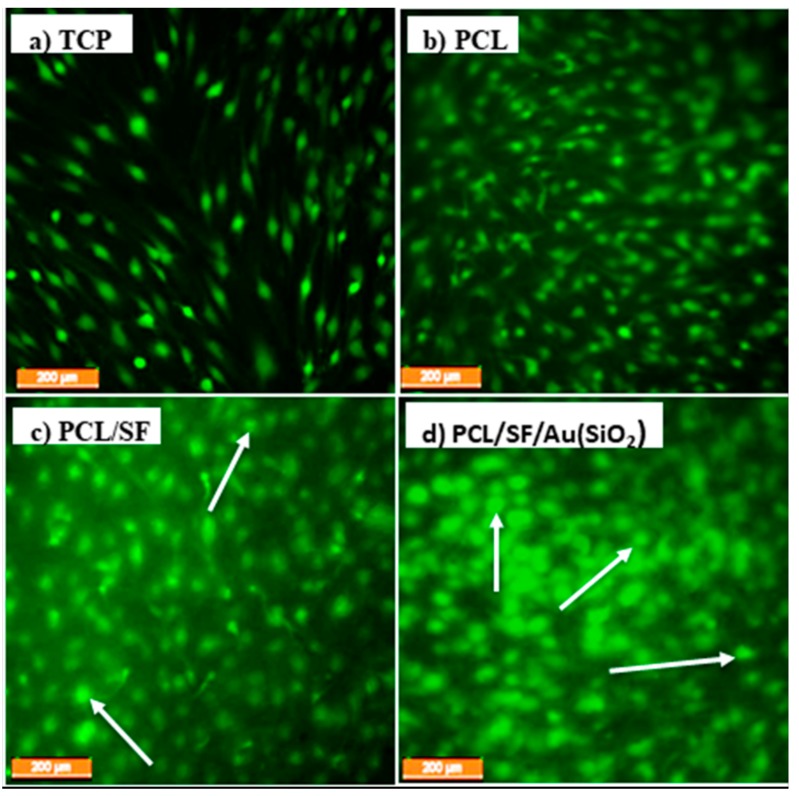
CMFDA dye extrusion image to analyze the cell morphology on (**a**) TCP, (**b**) PCL, (**c**) PCL/SF. and (**d**) PCL/SF/Au(SiO_2_) nanofibrous scaffolds at 10× magnifications (Scale bar: 200 µm). Arrows indicate the osteoblast-like morphology of the differentiated hMSCs.

**Figure 8 ijms-20-05135-f008:**
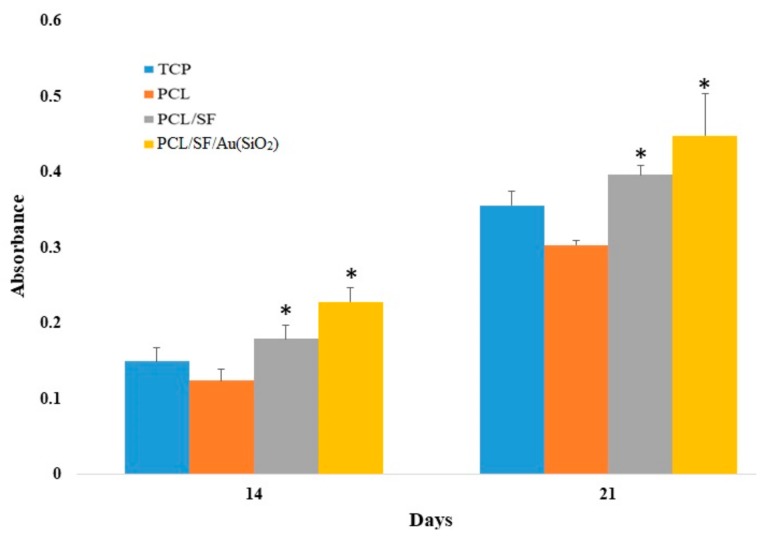
Alkaline phosphatase activity on TCP, PCL, PCL/SF and PCL/SF/Au(SiO_2_) nanofibrous scaffolds using hMSCs on day 14, and 21. * *p* < 0.05.

**Figure 9 ijms-20-05135-f009:**
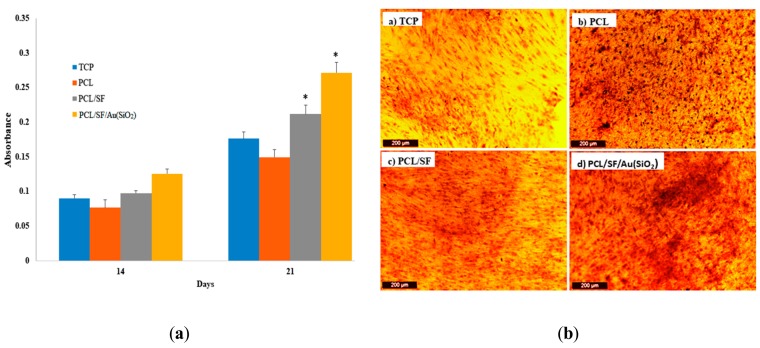
(**a**) Quantitative analysis of the mineralization by osteogenic differentiation of MSCs on the different scaffolds. (**b**) Optical microscope images showing the secretion of ECM by osteogenic differentiation of MSCs using Alizarin red staining on day 21 on a) TCP, b) PCL, c) PCL/SF and d) PCL/SF/Au(SiO_2_) nanofibrous scaffolds at 10× magnification (Scale bar: 200 µm). * *p* < 0.05.

**Figure 10 ijms-20-05135-f010:**
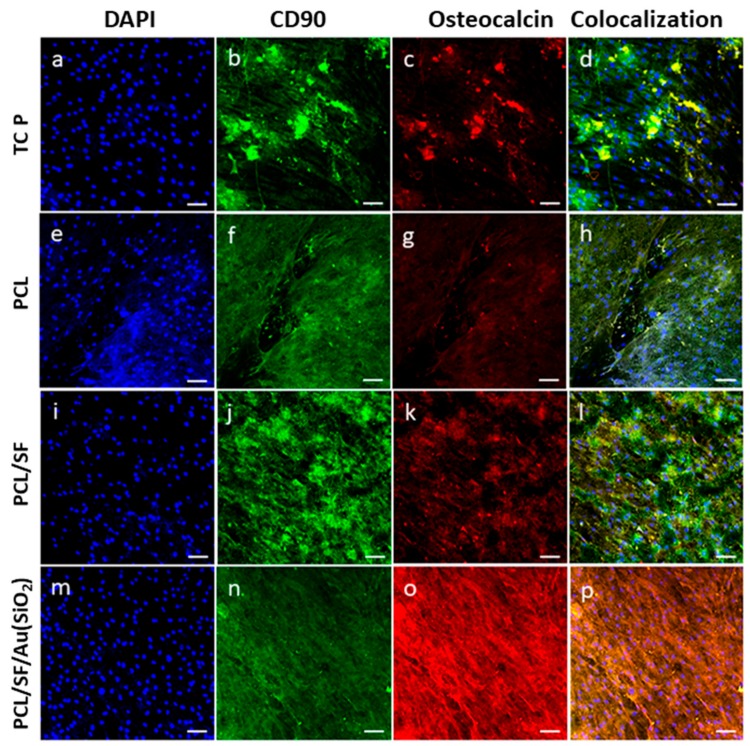
Confocal microscopy images to confirm osteogenic differentiation of MSCs using MSC specific marker protein CD90 (**b**,**f**,**j**,**n**) and osteoblast-specific marker protein osteocalcin (**c**,**g**,**k**,**o**). Merged image showing the dual expression of both CD90 and osteocalcin, characteristic of MSCs which have undergone osteogenic differentiation (**d**,**h**,**l**,**p**) on TCP, PCL, PCL/SF and PCL/SF/Au(SiO_2_) with the nuclear staining by DAPI (blue fluorescence). Nucleus stained with DAPI (**a**,**e**,**i**,**m**) at 20× magnification (Scale bar: 50 µm).

**Table 1 ijms-20-05135-t001:** Characterization of nanofibrous scaffolds.

Nanofibrous Constructs	Fiber Diameter (nm)	Pore Size (μm)	Porosity (%)	Tensile Strength (MPa)
PCL	215 ± 32.12	1.45 ± 0.26	88 ± 4.3	7.63
PCL/SF	164 ± 18.65	2.12 ± 0.31	92 ± 6.3	11.67
PCL/SF/Au(SiO_2_)	172 ± 24.22	2.35 ± 0.22	90 ± 7.5	12.11

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
