# Peer review of "Osteogenic Differentiation of Mesenchymal Stem Cells with Silica-Coated Gold Nanoparticles for Bone Tissue Engineering"

_ijms, 2019, doi:10.3390/ijms20205135_

Round 1

Reviewer 1 Report

The manuscript entitled “Osteogenic Differentiation of Mesenchymal Stem Cells with Silica Coated Gold Nanoparticles for Bone Tissue Engineering” is focused on an interesting topic. However, there are some points that the authors should clarify.

In the materials and methods section the authors reported that the scaffolds were characterized by SEM using JEOL 318 JSM-5600- LV-SEM, JEOL, Tokyo, Japan. However, they wrote that they used FESEM in the results section. Did the authors used SEM or FESEM?

The authors reported F-12 (HAM) in the material section (line 298) but then, they reported that the cells were cultured in DMEM. Did the author used DMEM-F-12 or DMEM alone?

The submission guidelines of the journal (https://www.mdpi.com/journal/ijms/instructions) specify that the manuscript sections should be: Introduction, Results, Discussion, Materials and Methods, Conclusions (optional). Therefore, the authors should separate the Results from the Discussion.

Figure 5 is inverted with figure 6. Figure 6 should be figure 5 and the legends should be fixed.

The p-value reported in figure 8 is not clear. The authors reported asterisks but they did not specify the comparisons. Did the authors find a statistical difference comparing PCL/SF with both TCP and PLC alone? Since the standard deviation is high, it is strange that they find a difference using ANOVA test. Which kind of post-hoc was used? The authors should specify the post-hoc used in the statistical paragraph as well as the software used for the analysis.

The images of the immunofluorescence are not clear (figure 10). It will be better to use an higher magnification and also to provide higher quality of the images.

Minor comments

The authors should use the same template for each graph of the figure (see figure 2, figure 10).

The authors should correct spaces and typos throughout the manuscript. For example: figure 1 PLACL/SF/Ai(SiO2) should be PCL/SF/Ai(SiO2); figure 2 PLACL/SF/Ai(SiO2) should be PCL/SF/Ai(SiO2); figure 4 PLACL/SF/Ai(SiO2) should be PCL/SF/Ai(SiO2); line 95 gron should be grown; line 139 H2O should be H2O etc.

All the abbreviation should be defined, for example line 304: HIFP.

Author Response

We are very much grateful to the reviewers for their indepth and thorough review. We have revised our paper in light of their useful suggestions and comments. We hope our revision has improved the paper to a level of their satisfaction.  Please find the attached. Thank you

Reviewer 2 Report

The manuscript describes a study testing the effects of incorporating silica and gold nanoparticles into PCL nanofiber scaffolds on osteogenesis of MSCs.

Is it realistic to create an electrospun scaffold of sufficient thickness for bone substitution?

The result and discussion section was combined. However, the data are not thoroughly enough discussed for my opinion. Particularly, limitations should be adressed.

The particular mode of action of the nanoparticles (hypothesis for a candidate signaling pathway stimulated) on the MSCs is not adressed

The degradabilty of the scaffolds remains not sufficiently adressed. Gold is not degradable, fibroin might also be stable, PCL degrades. How about the silica?

Dimension of the scaffold should be mentioned in more detail. Were for all experiments 15 mm diameter scaffolds used?, thickness? number of layers? Pore size and interconnectivity (line 20, abstract, line 105) should really be mentioned in the abstract. Pore size is important in regard to future vascularization (see abstract: propensity for neovascularization) which requires larger pores. The small pore size (table 1) should be discussed. which strategies could be applied in future to increase pore size?

However, there are many inconsistencies such as surplus blanks, lacking scale bars in many figures (e.g. Fig. 5, 7, 9, 10) which should be adapted. Sometimes there is a blank before the citation, sometimes not...or a blank between number and unit or not (163 and 164).

2.4. biomechanics: why was tensile strength assessed and not resistance to pressure? Values should be related to bone.

Detailed comments

abstract

line 18: the brackets should be completed

line 28: the explanation of the abbreviation CMFDA is not correct

remove all  surplus blanks throughout the whole manuscript

introduction

line 70: adapt font size/type

line 72: should the biomechanical properties of PCL be mentioned here?

line 80: "increases mineralisation by ... ca++ phosphates..." does it mean deposition of hydroxyapatite?

line 88: bring the citations together in one bracket, see also line 180

line 95: write correctly "grown on these"

line 99: instead of "generally" write "often"

results and discussion

2.1. describe thickness and overall dimensions, use the similar writing for fiber vs. fibre throughout the manuscript, e.g. line 112

table 1: is the difference between fiber diameter significant?

Figure 5: the legend is not correct. what mean the white and red arrows? Cell distribution and morphology is shown. Add the correct method.

figure 3: include TCP?

line 205: Extracellular matrix has already been abbreviated, refer to figure 5 instead of figure 6 here, similar line 256

line 206: indicated ECM deposition in the figure using arrows or asterisks

line 211: what is bioapatite? hydroxyapatide?

line 214: sentence is incomplete an "is" is lacking

Fig. 6: "Sio2" should be written "SiO2". The legend is not correct here.

line 236: write "results"

Fig. 7: morphology is not detectable in this figure, because the fluorescence overflows. Provide a sharper set of images.

Line 263: surplus point

Line 264: write scaffolds (plural)

line 282: CD90 is expressed by many mesenchymal cell types and not a real hMSC-specific marker as suggested here

Fig. 10: adapt the distances between singular images within the panel

methods

3.1. where hMSC donors pooled? at which passage were the cells used?

composition of especially this silica should be detailed here

line 332 rpm: calculate "g"

TEM was done to depict the fibers - I could not find the methods description

3.10: triplicates, were indeed independent experiments performed?

line 409-409: The most ideal surface, better to write „the most suitable surface among those tested“...

line 413: penetrate deep into the layers: depth of penetration should be measured

Author Response

(The authors gave the same response as above.)

Round 2

Reviewer 2 Report

The manuscript has been revised, however a many typical and grammar errors have to be corrected (examples below).

Legends of figure 7, 9 and 10: please mention the size of the novel inserted scale bar. „Fold magnification“ provides no substantial information and can be omitted. I would recommend separation of results and discussion part since I believe the data are not discussed deeply enough.

Minor:

Line 55: „new tissue formation has  takes place“ correct: has taken place?

Lines 60-63: „The combination of  scaffold and cells allows for  stimulation  of biomolecules like growth factors, angiogenic  factors which  stimulates cell function and tissue regeneration and scaffold eventually degrades and are replaced  by newly  grown tissue“ his sentence should be improved

Line 125: „results“ instead of „result“?

Line 153: „Interconnectivity of the pores and pore size are important factor in fabrication of scaffolds for tissue engineering“ better to write „factors“.

„shows  1.45μm,2.12μm“ insert a blank behind the coma „in to“ write: „into“

Line 270: „Calcium mineralization“ better: „calcium deposition“

„Further  calcium  mineralization  qualitatively  by  microcsopicmicroscopic  images.“ Write a complete sentence

Line 272. „ incorporation of Au(SiO2) upregulate“ better: „upregulates“

„Mineralization of MSC is one of the important factor for BTE“ „ECM mineralization by osteogenically differentiated MSCs is one of the most important factors for BTE“?

Author Response

We thank the reviewers for their helpful comments and suggestions. The above-mentioned paper has been strengthened after amendments made according to the comments of the reviewers, and we hope that this paper is now acceptable for publication in International Journal of Molecular Sciences.
